# Casein Lactose-Glycation of the Maillard-Type Attenuates the Anti-Inflammatory Potential of Casein Hydrolysate to IEC-6 Cells with Lipopolysaccharide Stimulation

**DOI:** 10.3390/nu14235067

**Published:** 2022-11-29

**Authors:** Na Chen, Yu Fu, Zhen-Xing Wang, Xin-Huai Zhao

**Affiliations:** 1Maoming Branch, Guangdong Laboratory for Lingnan Modern Agriculture, Guangdong University of Petrochemical Technology, Maoming 525000, China; 2School of Biological and Food Engineering, Guangdong University of Petrochemical Technology, Maoming 525000, China; 3Research Centre of Food Nutrition and Human Healthcare, Guangdong University of Petrochemical, Maoming 525000, China; 4College of Food Science, Southwest University, Chongqing 400715, China

**Keywords:** anti-inflammatory activity, casein, glycation, lactose, Maillard reaction

## Abstract

During the thermal processing of dairy products, the Maillard reaction occurs between milk proteins and lactose, resulting in the formation of various products including glycated proteins. In this study, lactose-glycated casein was generated through the Maillard reaction between casein and lactose and then hydrolyzed by a trypsin preparation. The anti-inflammatory effect of the resultant glycated casein hydrolysate (GCH) was investigated using the lipopolysaccharide (LPS)-sitmulated rat intestinal epithelial (IEC-6) cells as a cell model and corresponding casein hydrolysate (CH) as a control. The results indicated that the preformed glycation enabled lactose conjugation to casein, which endowed GCH with a lactose content of 12.61 g/kg protein together with a lower activity than CH to enhance the viability value of the IEC-6 cells. The cells with LPS stimulation showed significant inflammatory responses, while a pre-treatment of the cells with GCH before LPS stimulation consistently led to a decreased secretion of three pro-inflammatory mediators, namely, IL-6, IL-1β and tumor necrosis factor-α (TNF-α) but an increased secretion of two anti-inflammatory mediators, including IL-10 and transforming growth factor-β (TGF-β), demonstrating the anti-inflammatory potential of GCH in LPS-stimulated cells. In addition, GCH up-regulated the expression of TLR4, p-p38, and p-p65 proteins in the stimulated cells, resulting in the suppression of NF-κB and MAPK signaling pathways. Collectively, GCH was mostly less efficient than CH to exert these assessed anti-inflammatory activities in the cells and more importantly, GCH also showed an ability to cause cell inflammation by promoting IL-6 secretion and up-regulating the expression of TLR4 and p-p65. The casein lactose-glycation of the Maillard-type was thereby concluded to attenuate the anti-inflammatory potential of the resultant casein hydrolysate. It is highlighted that the casein lactose-glycation of the Maillard-type might cause a negative impact on the bioactivity of casein in the intestine, because the glycated casein after digestion could release GCH with reduced anti-inflammatory activity.

## 1. Introduction

As the main site of food digestion and nutrient absorption, the function of the intestine depends on its mucosal epithelial barrier [1]. The intestinal barrier, formed by the tight junctions of small intestinal epithelial cells (IECs), is important for maintaining the homeostasis of the internal environment of the intestine. However, the intestine is often in an inflammatory state due to the prolonged exposure to the harmful intestinal microorganisms and some toxic substances, while this condition might be linked to several chronic illnesses including cancer and inflammatory bowel disease (IBD) [2]. The well-known Gram-negative bacteria *Escherichia coli* are the common marker species of intestinal inflammation, because lipopolysaccharide (LPS), a major component of the cell wall, can cause both food poisoning and inflammatory intestinal lesions [3,4]. It is known that LPS can easily damage IECs, resulting in these adverse events, such as reduced barrier function, increased permeability, intestinal inflammation, and abnormal immune reactions [5]. In response to LPS damage or infections, IECs will release inflammatory mediators; meanwhile, the intrinsic immune cells, such as macrophages and dendritic cells, will identify the receptors, bind to pathogens or toxins, and start an inflammatory cascade response [6]. In addition, the regulatory T lymphocytes (Treg) maintain the homeostasis by suppressing the aberrant immune responses against microbiota or dietary antigens [7]. Overall, IECs display a critical role in preventing internal exposure to the harmful microorganisms and toxins. Thus, it is particularly important to reduce intestinal inflammation by mitigating IEC damage induced by LPS and other substances [8]. Fortunately, several food components, including polysaccharides, polyphenols, proteins, and other bioactive compounds, have a helpful modulatory effect on the induced intestinal inflammation.

In general, these food compounds are considered to have the ability to control the release of inflammatory cytokines from stimulated model cells (e.g., Caco-2, RAW 264.7, and IEC-6 cells) [9], or affect intestinal morphology and induce inflammatory signs and symptoms to alleviate intestinal inflammation [10]. It was reported that the polysaccharides from *Ganoderma atrum* ((Curtis) P. Karst.) had an anti-inflammatory effect on the LPS-induced Caco-2/RAW 264.7 co-cultured cells via reducing the secretion of inflammatory cytokines and activating the MAPK signaling pathway [11], while the polysaccharides from *Arctium lappa* modulated several inflammatory and anti-inflammatory cytokines to inhibit DSS-induced colitis in mice [12]. Growing evidence suggests that polyphenols in the intestine also have an anti-inflammatory function. Mango polyphenols were reported to decrease the secretion of inflammatory cytokines but increase the secretion of anti-inflammatory cytokines in inflammatory Caco-2 and HIEC-6 cells; meanwhile, they showed an ability to inhibit the NF-κB pathway [13]. In DSS-induced colitis in rats, propolis polyphenols were capable of inhibiting the secretion of inflammatory markers effectively; subsequently, the colitis index and colonic apoptosis were reduced efficiently [14]. Furthermore, several proteins, or their hydrolysates, also possess anti-inflammatory activities. It has been shown that when egg white was subjected to a simulated digestion, the resultant hydrolysate displayed anti-inflammatory activity via regulating the expression of cytokines that either promoted or inhibited cell inflammation [9]. Tenore and coauthors used a model of DSS-induced colitis to examine the anti-inflammatory activity of buffalo milk, and reported that buffalo milk could reduce clinical symptoms including intestinal atrophy and intestinal barrier damage, and modulate the NF-κB and MAPK pathways [15]. It is well-agreed that casein can react with lactose during dairy processing, because of the inevitable Maillard reactions. Chemically, the free NH_2_ of proteins (primarily the ε-NH_2_ in Lys) and the carbonyl groups of reducing saccharides are what typically cause the Maillard reaction of proteins and reducing saccharides. More importantly, at the early stage of this reaction, casein can be glycated by lactose, which leads to the formation of the lactose-glycated casein. However, whether the glycated casein has different biofunctions than the original casein in the intestine is still unknown.

The Maillard reaction has been verified to impact several physicochemical properties of proteins such as solubility, thermal stability, emulsification, and viscosity [16]. Meanwhile, recent studies have also reported that this reaction causes change in these biofunctions, including anti-oxidant, anti-bacterial, and anti-hypotensive effects [17,18,19]. Additionally, it was evidenced that the Maillard product of a gelatin hydrolysate and galactose showed an inhibitory effect on the inflammation of macrophages [20], while the glycated proteins of the Maillard-type possessed a probiotic effect in the gastrointestinal tract [21]. The Maillard reaction frequently produces what is termed advanced glycation end products (AGEs) [22], and the foods with high AGEs levels might be adverse to human health. For example, Hillman and coauthors found that if young rats were fed with skimmed milk powder containing high levels of AGEs, the rats showed the development of intestinal inflammation and immune system disorders [23]. In addition, the AGEs from baked foods could be accumulated in the intestinal tissues of the rats and older mice, and subsequently induce intestinal inflammation or other diseases [24,25]. Furthermore, it was also observed that dietary AGEs could encourage cellular inflammation and trigger the release of the inflammatory cytokines, namely TNF-α, in human macrophages both before and after digestion [26]. In one of our previous studies, it was highlighted that a casein lactose-glycation of the Maillard-type caused a lower barrier enhancement for the casein hydrolysate in rat small intestinal epithelial (IEC-6) cells [27], indicating an adverse event of this reaction on the bioactivities of casein in the intestine. Based on these past results, it is thus inferred that this casein lactose-glycation, as one of the early reactions of the Maillard reaction, might have an unrevealed impact on the anti-inflammatory effect of the resultant glycated casein hydrolysates in the intestine, because the lactose-glycated casein in intestine will be enzymatic hydrolyzed. Clearly, possible activity changes induced by this casein glycation is vital to the healthcare function of casein or dairy products in the intestine. Thus, such an investigation deserves our consideration.

In this study, casein was glycated with lactose via the Maillard reaction and then hydrolyzed with a trypsin preparation to yield the targeted glycated casein hydrolysate (GCH). After this, utilizing the LPS-stimulated IEC-6 cells as a cell model and the unglycated casein hydrolysate (CH) as a control, the anti-inflammatory potential of GCH was assessed systemically. In brief, the two hydrolysate samples were assessed and compared for their activities to regulate cell viability, cytokine secretion, and expression of several proteins involved in the TLR4/p38 MAPK/NF-κB signaling pathway. The aim of this study was to reveal whether this casein glycation of the Maillard-type could cause a positive or negative change in the intestine, regarding the capacity of the resultant casein hydrolysate to attenuate the LPS-induced IEC inflammation.

## 2. Materials and Methods

### 2.1. Regents and Materials

Casein (sodium salt, protein content of 911.2 g/kg on dry basis), lactose, 3-(4,5-dimethyl-2-thiazolyl)-2,5-diphenyltetrazolium bromide (MTT), Dulbecco’s modified Eagle’s medium (DMEM), and lipopolysaccharide (LPS, produced from *E. coli* O55:B5) were all purchased from Sigma-Aldrich Chemical Co. (St. Louis, MO, USA). Trypsin-EDTA (ethylenediamine tetra-acetic acid), phosphate-buffered saline (PBS), Dimethyl sulfoxide (DMSO) and bovine insulin were given by Solarbio Technology Ltd. (Beijing, China), while fetal bovine serum (FBS) was purchased from Wisent Inc. (Montreal, QC, Canada). The trypsin (EC 3.4.21.4) preparation (120 kU/g) was the procured of Beijing Auboxing Biotechnology Co. (Beijing, China), and other chemicals utilized were of analytical grade.

The EnzyChromTM Galactose Assay Kit was supplied by BioAssay Systems (Hayward, CA, USA), while the enzyme-linked immunosorbent assay (ELISA) kits used to assay tumor necrosis factor (TNF), transforming growth factor (TGF), and three interleukins (IL-1β, IL-6, and IL-10) were all purchased from Nanjing Jiancheng Institute of Biological Engineering (Nanjing, China). Beyotime Biotechnology Institute (Shanghai, China) provided the BCA protein assay kit and radio immunoprecipitation assay (RIPA) lysis buffer. The phosphorylated p38 MAPK (#4511) and goat anti-rabbit HRP secondary antibody (#7074) were purchased from Cell Signaling Technology (Danvers, MA, USA), while the primary antibodies namely p-p65 (phospho-NF-B p65, Bioss bs-0982R) and TLR4 (Bioss bs-20594R), as well as the endogenous standard β-actin (Bioss bs-0061R) were purchased from Biosynthesis Biotechnology Inc. (Beijing, China).

### 2.2. Sample Preparation

The lactose-glycated casein was prepared as previously described [28]. Briefly, the casein solution (50 g protein per liter) was mixed fully with a lactose solution of 80 g/L to reach a fixed mass ratio of 1:1.6 and then adjusted to pH 6.8. The mixture was heated at 100 °C for 3 h, cooled to 20 °C, and adjusted to pH 4.5. To obtain the lactose-glycated casein, the precipitate fraction was centrifuged (5000× *g*, 20 min), washed twice with water of pH 4.5, neutralized to pH 7.0 using 0.5 mol/L NaOH, and then freeze-dried. Meanwhile, casein was heated at 100 °C for 3 h, cooled to 20 °C, added with lactose at the fixed mass ratio of 1:1.6, and then adjusted to pH 4.5. The precipitate fraction generated was treated similarly to obtain the control casein.

The control casein and glycated casein were dispersed in water to reach pH 7.0 and a protein content of 50 g protein per liter. The targeted protein hydrolysis was carried out with the addition of trypsin (7 kU/g protein) at 37 °C for 4 h, while this hydrolysis was stopped by heating at 100 °C for 5 min. Following a cooling to 20 °C and pH adjustment to 4.5, the hydrolysis solutions were centrifuged at 5000× *g* for 20 min to collect the supernatants, which were lyophilized to produce the respective CH and GCH.

### 2.3. Assays of Protein and Lactose Content as Well as the Degree of Hydrolysis

Protein content was measured using the Kjeldahl method with a conversion factor of 6.38 [29]. The two hydrolysates were hydrolyzed with 4 mol/L trifluoroacetic acid at 100 °C for 4 h, cooled to 20 °C, neutralized to pH 7.0 using 0.5 mol/L NaOH, and then diluted into a fixed volume of 25 mL. As previously described [28], the diluted solutions were detected for galactose content using the Galactose Assay Kit, while the lactose content (g/kg protein) of the hydrolysates was calculated using the measured galactose content and chemical formulae of lactose and galactose. The free amino groups of the hydrolysates were measured as previously described [30], while the degree of hydrolysis (DH) was thus calculated accordingly [31].

### 2.4. Cell Line and Cell Culture

Briefly, IEC-6 cells were grown in accordance with the instruction of the cell supplier, using an incubator set at 37 °C, a humidified environment of 5% CO_2,_ and the complete medium DMEM media supplemented with 4.5 g/L glucose, 4 mmol/L L-glutamine, 1.5 g/L NaHCO_3_, 1 mmol/L sodium pyruvate, and 10% fetal bovine serum. Twice per week, the medium was replaced. The experiments for this study used the cells that had fused to an average of 80%.

### 2.5. Assay of Cell Viability with or without LPS Exposure

The MTT assay was used to gauge the viability values of the treated cells. In brief, 200 μL of IEC-6 cells were cultured in 96-well plates at 1 × 10^4^ cells/well for 24 h, serum-starved for 12 h, and then incubated with or without CH/GCH at doses of 25–100 μg/mL for 12–48 h. Each well received 200 μL of MTT solution after the liquid had been removed, then incubated for 4 h. After discarding the supernatant, 150 μL of dimethyl sulfoxide was added to ensure that the generated formalin crystals were completely dissolved. Optical density at 450 nm was detected by a microplate reader (Bio-Rad Laboratories, Hercules, CA, USA). Viability values were calculated as previously explained [32], while the control cells without CH/GCH treatment were represented with a viability value of 100%.

Otherwise, IEC-6 cells at 1 × 10^4^ cells/well were cultured in 96-well plates for 24 h, incubated for 12 h in serum-starved media, incubated with or without CH/GCH at 25–50 μg/mL for 12 and 24 h, and then exposed to 10 μg/mL LPS for 24 h. Afterwards, the cells were treated as above to measure their viability values.

### 2.6. Assay of Cytokine Secretion

Two milliliters of cells were loaded into 6-well plates (1 × 10^5^ cells/mL) for 24 h, and then incubated in serum-free medium for 12 h. Following the removal of medium, the cells were treated with medium, CH or GCH (doses of 25–50 μg/mL) for 12–24 h, followed by LPS exposure (10 μg/mL) for 24 h. After these treatments, the supernatants were collected by centrifugation at 500× *g* for 20 min, while the concentrations of the three inflammatory mediators (IL-1β, IL-6, and TNF-α) and two anti-inflammatory mediators (IL-10 and TGF-β) were measured using the respective ELISA kits.

### 2.7. Assay of Protein Expression

In a nutshell, 5 mL of the cells were grown in the cell flasks for 24 h, treated with serum-free media for 12 h, incubated with or without CH and GCH at a dose of 25 μg/mL for 24 h, and then exposed to 10 μg/mL of LPS for 24 h. The cells were removed from the media, washed twice with a PBS (10 mmol/L, pH 7.4), added with 100 μL of the RIPA lysis buffer with protease inhibitor and PMSF (1 mmol/L), lysed on ice for 0.5 h, and then centrifuged at 12,000× *g* for 5 min at 4 °C. Using the BCA protein analysis kit, the obtained supernatants were evaluated for protein content. On a 12% SDS-PAGE gel, the same amount of proteins (20 μg) were separated and subsequently electro-transferred onto nitrocellulose membranes. The membranes were placed in 5% skimmed milk in TBST (1 × TPBS containing 0.1% Tween-20) for 2 h at 37 °C, and then kept overnight at 4 °C with the primary antibody (1:1000 dilution). The membranes were then washed three times in the TBST, exposed to peroxidase-conjugated secondary antibody (1:5000 dilution) at 37 °C for 2 h, and then rinsed in the TBST again. The immunolabeled proteins were identified using a 100 μL of enhanced chemiluminescence reagent (Fujifilm, Tokyo, Japan), while the protein blot pictures were obtained using an Image Quant LAS 500 (Fujifilm, Tokyo, Japan). Protein blots were analyzed quantitatively using the ImageJ software (National Institutes of Health, Bethesda, MA, USA), while β-actin was utilized as an endogenous standard to normalize the band density.

### 2.8. Statistical Analysis

All data collected was from at least three independent experiments or assays were reported as the mean values or mean values ± standard deviations. Significant differences (*p* < 0.05) between the mean values of different groups were analyzed by the IBM Statistical Products and Services Solutions (SPSS) 26.0 software (SPSS Inc., Chicago, IL, USA), using Duncan’s multiple comparison test and one-way analysis of variance (ANOVA).

## 3. Results

### 3.1. Effect of the Two Hydrolysates on the Growth Performance of IEC-6 Cells

The analysis results showed that the respective lactose content of the generated glycated casein and GCH were 14.89 and 12.61 g/kg protein, suggesting that the lactose molecules were attached to casein covalently via the Maillard reaction. Meanwhile, CH was found without detectable lactose content. Because GCH shared closed DH values with CH (14.2% versus 13.3%) but a different lactose content, this covalent lactose conjugation was thus regarded to give GCH different potential to CH in the cells. When GCH and CH were applied separately to the cells using the same dose and treatment time, the results from the MTT assay indicated that the treated cells possessed different viability values (Figure 1). In detail, when the cells were incubated with CH at 25–100 μg/mL for 12 h, the viability values were 106.5–122.8%; if the cells were treated with CH at 25–100 μg/mL for 24 h, the detected viability values ranged from 101.8% to 122.6%; moreover, the cell treatment of 48 h by CH led to viability values of 101.7–125.7%. The data demonstrated that CH at these doses had no cytotoxic effect on the cells. Meanwhile, when the cells were treated with GCH of 25–100 μg/mL for 12 h, the measured viability values ranged from 98.5% to 115.5%, and a longer treatment time of 24 or 48 h by GCH caused viability values of 100.5–115.8% (or 92.3–103.7%). Clearly, GCH was less potent than CH in the cells to promote cell growth. More importantly, GCH at a higher dose (i.e., 100 μg/mL) together with a longer treatment time (i.e., 48 h) showed a cytotoxic effect on the cells, yielding viability values of the treated cells near to 92%. This fact implied that the casein glycation inhibited the growth and proliferation of GCH in the cells, suggesting a negative effect of Maillard-type glycation on casein bioactivity in the intestine. In addition, based on the measured data, the two sample doses (25 and 50 μg/mL) and two treatment times (12 and 24 h) were selected in the later evaluations.

### 3.2. Effect of the Two Hydrolysates on the LPS-Induced Cellular Injury

To reveal the potential protection of GCH and CH on the cells to combat against LPS-induced damage, the cells were treated with doses of 25–50 μg/mL for 12–24 h and then subjected to 10 μg/mL LPS for 24 h. The assaying results showed that LPS caused undoubted cell injury, reflected by the decreased viability values (83.6–88.6%, Figure 2). Meanwhile, both CH and GCH alleviated LPS-induced injury through enhancing the viability of the treated cells (Figure 2). In detail, when the cells were cultured with CH and GCH for 12 h before the LPS injury, the measured viability values were increased to 97.5–99.1% and 95.8–97.3%, respectively; if the cells were cultured with CH and GCH for 24 h before LPS exposure, the respective viability values increased up to 95.8–97.3% and 89.8–93.9%. Clearly, CH showed a higher activity than GCH of the same dose in protecting the cells from LPS-induced injury, declaring that the lactose glycation reduced the activity of GCH to protect IECs from LPS-induced injury. In other words, casein lactose-glycation of the Maillard-type was regarded to decrease casein biofunction in the intestine to alleviate the toxic effect of LPS.

### 3.3. Effect of the Two Hydrolysates on the Secretion of Three Inflammatory Mediators

When IEC-6 cells were treated with CH or GCH separately at doses of 25–50 μg/mL for 12–24 h but without LPS injury, the secretion of one important inflammatory cytokine IL-6 was slightly or clearly enhanced (Figure 3). The results indicated that the control cells had IL-6 levels of 39.9–43.3 pg/mL (12–24 h), while the CH-treated cells had IL-6 levels of 42.5–49.2 pg/mL (12 h) or 41.6–46.4 pg/mL (24 h). However, the GCH-treated cells showed enhanced IL-6 levels of 56.6–68.1 pg/mL (12 h) or 49.7–51.4 pg/mL (24 h). This fact meant that GCH might potentially induce cellular inflammation via enhancing IL-6 secretion, which should be investigated further in future work.

Once the cells were treated with CH/GCH at doses of 25–50 μg/mL for 12–24 h and then exposed to 10 μg/mL LPS for 24 h, IL-6 secretion of the treated cells showed a response to the treatments (Figure 4A,B). Control cells, without LPS injury, were cultured for the same time periods as the model cells (12 and 24 h plus 24 h) and exhibited an IL-6 secretion of 90.6–92.2 pg/mL. In the model cells, IL-6 secretions were up to 173.4 pg/mL (12 h) and 168.5 pg/mL (24 h), confirming LPS-induced cell inflammation. Meanwhile, both CH and GCH showed an anti-inflammatory effect on the injured cells, because they efficiently inhibited IL-6 secretion in LPS-injured cells. To be more specific, CH reduced IL-6 secretion to 122.4–143.8 pg/mL (12 h) and 104.0–124.1 pg/mL (24 h), while GCH decreased IL-6 secretion to 159.4–173.2 pg/mL (12 h) and 146.6–160.8 pg/mL (24 h). Consistently, CH was more active than GCH in inhibiting IL-6 secretion. In addition, a CH/GCH dose of 25 μg/mL and a treatment time of 24 h, IL-6 secretions were even more efficiently reduced. Thus, it was concluded again that the assessed lactose glycation of casein might cause a weaker anti-inflammatory function for the casein hydrolysate in the intestine with LPS exposure.

Furthermore, when both IL-1β and TNF-α were included as evaluation indices (Figure 4C–F), it was found that both CH and GCH had an anti-inflammatory effect on the wounded cells by reducing IL-1β and TNF-α secretion. In detail, the control cells had the lowest IL-1β and TNF-α secretion (6.5–8.5 and 80.0–86.0 pg/mL), while the model cells had the highest (33.0–35.2 and 138.8–148.8 pg/mL). When the cells were treated with CH before LPS injury, IL-1β secretion was 21.8–23.0 pg/mL (12 h) and 21.1–23.9 pg/mL (24 h), while TNF-α secretion was 107.3–111.9 pg/mL (12 h) and 87.8–96.3 pg/mL (24 h). If the cells were treated with GCH before LPS injury, the measured IL-1β secretion was 29.2–32.4 pg/mL (12 h) and 24.0–23.7 pg/mL (24 h), and the detected TNF-α secretion was 113.6–126.0 pg/mL (12 h) and 105.3–123.3 pg/mL (24 h). Overall, using a sample dose of 25 μg/mL and a cell treatment of 24 h led to a reduction in IL-1β/TNF-α secretion, with CH being more potent than GCH in reducing IL-1β/TNF-α secretion. The performed lactose glycation of casein was thus regarded to cause a negative effect on the anti-inflammatory activity of the resultant casein hydrolysate, based on the obtained data.

### 3.4. Effect of the Two Hydrolysates on the Secretion of Two Anti-Inflammatory Mediators

When IL-10 and TGF-β were utilized as two assessment indicators, it was also discovered that both CH and GCH had an anti-inflammatory effect on LPS-injured cells (Figure 5). The control cells had the highest IL-10 and TGF-β secretion (21.3 and 47.4 pg/mL in 12 h, or 22.2 and 39.7 pg/mL in 24 h), whereas the model cells showed the lowest IL-10 and TGF-β secretion (6.7 and 18.8 pg/mL in 12 h, or 8.1 and 15.4 pg/mL in 24 h) as a result of LPS-induced cell inflammation. When the cells were treated with CH for 12 h before LPS injury, the respective IL-10 and TGF-β secretion increased to 13.4–15.2 and 16.7–17.0 pg/mL; if the cells were treated with CH for 24 h, IL-10 and TGF-β secretion increased to 16.9–17.2 and 34.0–34.1 pg/mL, respectively. Meanwhile, GCH-treated cells showed a IL-10 and TGF-β secretion of 7.5–10.3 and 14.2–15.1 pg/mL (12 h), and 9.5–12.0 and 30.0–30.1 pg/mL (24 h), respectively. Both CH and GCH had a capacity to elevate the secretion of the two anti-inflammatory cytokines, suggesting their anti-inflammatory effect on the cells once more. In addition, the CH/GCH dose of 25 μg/mL and cell treatment of 24 h were more efficient in increasing IL-10 and TGF-β secretion, and CH showed a higher potential than GCH in enhancing IL-10 and TGF-β secretion. Given that GCH caused less IL-10 and TGF-β secretion, it was proven again that the lactose glycation of casein had a detrimental impact on casein bioactivity in the intestine.

### 3.5. Expression Changes in Signaling Pathway-Related Proteins in IEC-6 Cells

Three cellular proteins, namely, TLR4, p-p38, and p-p65, which are crucial to the TLR4/p38 MAPK/NF-κB signaling pathway, were examined for their expression levels in the cells to further support the anti-inflammatory action of CH and GCH in LPS-stimulated cells (Figure 6). TLR4:β-actin, p-p65:β-actin, and p-p38:β-actin ratios were also computed as normal (Table 1). The results indicated that the applied LPS injury led to the up-regulated expression of TLR4, p-p65, and p-p38 in the model cells. Meanwhile, CH showed an ability to inhibit the up-regulation of the three proteins in the inflammatory cells, resulting in lower expression levels of the three proteins. GCH was also detected with an activity (but weaker than CH) to inhibit the p-p38 up-regulation in the inflammatory cells. However, GCH was also detected to possess a capacity to promote TLR4 and p-p65 expression. That is, GCH itself was able to induce cell inflammation by up-regulating the expression of p-p65 and TLR4. Thus, the lactose glycation of casein might bring about a negative effect for the resultant casein hydrolysate, highlighted by its unrevealed potential to cause IEC inflammation and a weaker ability to protect IEC from LPS-induced IEC inflammation. This finding might be critical in dairy products with heat treatment and thus should be efficiently clarified in future studies.

## 4. Discussion

The protein glycation of the Maillard- or transglutaminase (TGase)-types are two ways that can modify protein properties effectively. Previous studies have confirmed that the TGase-induced protein glycation could greatly improve the functional properties and bioactivities of proteins [33]. Meanwhile, another method to glycate food proteins is the Maillard reaction, which is observed during the thermal processing of food via the reaction of proteins with reducing saccharides. Chemically, protein glycation of Maillard-type occurs in the early stage of the Maillard reaction. Past results have confirmed that the Maillard-type lactose glycation in casein occurs on the lysine residues of α_S1_-casein, β-casein, and κ-casein, while the imported amount of lactose in caseins was about 10.58 g/kg protein [34,35]. Maillard-type glycation thus can alter the protein properties, such as rheological behaviors, foaming property, thermal stability, gelation, and others [36]. For example, the water-solubility and emulsification of buckwheat proteins were improved by conjugating polysaccharides on to the surface of protein molecules via Maillard reactions, while the antigenicity of the buckwheat proteins could also be reduced by this glycation [37]. In addition, the products from Maillard reactions have been verified to possess a variety of bioactivities. It has been observed that the anti-oxidant, anti-bacterial, and anti-inflammatory activities of glycated proteins are enhanced; however, their anti-irritability is reduced [38]. When β-lactoglobulin was glycated with soluble chitosan by Maillard reactions, the glycated proteins showed a better power to reduce ferric ions, suggesting an enhanced anti-oxidative activity [39]. When egg white lysozymes were glycated by guar gum via Maillard reactions, the products detected were able to inhibit Gram-negative and Gram-positive bacteria [40].

Unfortunately, Maillard reactions are also regarded to have some adverse effects on proteins. For example, these reactions cause the loss of some amino acids, especially the essential amino acid lysine, because this reaction prevents lysine from being recognized by hydrolytic enzymes in the digestive tract and thus lead to a reduction in protein digestibility [41]. Maillard reaction products thus have lower absorption and bio-availability in vivo [42,43], while casein reacted with glucose and digested by trypsin and digestive enzymes was also found to have a decreased anti-oxidative activity [44]. It was also proven in the previous three studies of our group that casein lactose-glycation of Maillard-type was detrimental to the in vitro activities of the resultant casein hydrolysate, because this hydrolysate exerted a decreased barrier enhancement on IEC-6 cells [27], a lower growth proliferation and anti-apoptotic effect on IEC-6 cells [32], and a reduced immunomodulatory effect on the targeted immune cells [45]. Furthermore, the in vivo results from weaned rats also demonstrated that the casein lactose-glycation of the Maillard-type impaired the ability of casein hydrolysates to promote the development of small intestine and secretion of three digestive enzymes or three brush-border enzymes [46]. Our daily diets usually contain various products generated from Maillard reactions (especially AGEs). The consumption of these products is thus inevitable, while the potential effect of these products on the body must be clarified. For example, it was proven that dietary AGEs before and after digestion have an ability to promote cellular inflammation in human macrophages [26]. Along with the previously described investigations, the current study confirmed that the performed lactose glycation of casein reduced the ability of the resultant casein hydrolysate to alleviate LPS-induced cellular inflammation (Figure 4 and Figure 5). Additionally, and possibly more crucial, the resultant casein hydrolysate itself might have the ability to trigger cell inflammation by encouraging the release of the pro-inflammatory mediator IL-6 and activating the proteins TLR4 and p-p65 (Figure 3 and Table 1). In addition, the resultant casein hydrolysates at a higher dose (i.e., 100 μg/mL) also showed a cytotoxic effect on the cells (Figure 1). All these results consistently demonstrated a negative effect of the assessed casein lactose-glycation on casein bioactivity in the intestine.

When the body is stimulated by hazardous chemicals, the immune system causes a sequence of physiological reactions known as inflammation [5]. The release of LPS from pathogenic bacteria in the gut can directly cause local irritation of the intestine resulting in significantly increased levels of cytokines, such as IL-1β, IL-6, and TNF-α, together with reduced levels of anti-inflammatory cytokines, such as IL-10 and TGF-β [6,47]. However, in normal organisms the secretion of these inflammatory and anti-inflammatory cytokines is in a dynamic balance. It is known that the inflammatory inducers, such as LPS, activate inflammatory mediators existing in the body and then trigger inflammatory responses through a variety of signaling pathways [48]; for example, both the NF-κB and MAPK signaling pathways are involved in cellular inflammation. The NF-κB signaling pathway is related to the transcriptional regulation of genes involved in the inflammatory response [49], while the MAPK signaling pathway is involved in the inflammatory response of the body through a phosphorylation cascade [50]. In addition, the Toll-like receptor 4 (TLR4) is also associated with LPS-induced signal transduction. When LPS is recognized by the TLR4 receptor, both NF-κB and MAPK signaling pathways are thus activated in the cells, while the expression of proteins such as TLR4, p65, IκBα, p38, and JNK are up-regulated, which in turn triggers cytokine secretion [51,52]. Food intake causes direct contact between food components and the intestine, which thus ensures food components to directly display their anti-inflammatory activities to IECs. It was reported that acid hydrolysis of pepper pectin polysaccharides yielded a polysaccharide fragment (CA-H), which had a mitigating effect on cell inflammation via reducing TNF-α release and increasing IL-10 production in a LPS-stimulated mouse model [53]. It was also found that naringin fractions showed an anti-inflammatory effect on LPS-stimulated RAW 264.7 cells by inhibiting the expression of iNOS and COX-2 but reducing the secretion of IL-1β and TNF-α [54], while soy proteins displayed anti-inflammatory potential in mature mouse adipocytes by suppressing the secretion of the three cytokines, TNF-α, MCP-1, and IL-6 [55]. Casein as a component of milk proteins is degraded into casein hydrolysate when entering the intestine, where casein hydrolysates can exert their bioactivities on IECs. Therefore, GCH and CH (but not glycated casein or unglycated casein) were used to assess their anti-inflammatory effect on LPS-stimulated cells. Similar to results of the aforementioned studies, the anti-inflammatory activities of the assessed hydrolysates on the stimulated cells were verified or proven in this study. However, current findings suggested that a prior lactose-glycation of casein via Maillard reactions before casein digestion could decrease the anti-inflammatory effect of the resultant casein hydrolysate in the intestine, highlighting an adverse effect of Maillard-type protein glycation on casein bioactivity.

The effect of proteins and their hydrolysates on intestinal inflammation has received extensive attention. A bioactive peptide (KQS-1) generated from the digestive process of extruded sialoprotein was reported to fight against the inflammation caused by LPS in macrophages through inhibiting the activation of the NF-κB signaling pathway and lowering the expression of IL-1β, IL-6, TNF-α, and MCP-1 [56]. In LPS-treated IEC-6, bovine lactoferrin also had an anti-inflammatory effect by reducing the secretion of IL-6, TNF-α, and IL-1β, as well as suppressing MAPK activation and the nuclear translocation of NF-κB [57]. When casein hydrolysates were modified by plastein reactions in the presence of exogenous amino acids, the obtained modifiers demonstrated greater anti-inflammatory activity against LPS-induced macrophage inflammation, mostly by controlling IL-6 and TNF-α production and preventing the activation of the NF-κB signaling pathway [58]. However, certain protein by-products generated during the Maillard process seem to have an ability to encourage the undesirable cellular inflammatory response. For example, it has been observed that the pro-inflammatory cytokine TNF-α is secreted in RAW 264.7 macrophages in response to the dietary AGEs [23]. Moreover, AGEs in skimmed milk powder was shown to cause intestinal inflammation in young rats [26], and the secretion of the inflammatory cytokine TNF-α in human macrophages was enhanced by dietary AGEs [59]. Sharing a conclusion consistent with the aforementioned studies, this study found that CH in LPS-stimulated cells had an anti-inflammatory effect; however, due to casein glycation of the Maillard-type, GCH showed a lower anti-inflammatory effect than CH and also promoted IL-6 secretion and up-regulation of TLR4/p-p65 expression. Thus, casein lactose-glycation as an early reaction of the Maillard reaction was regarded to induce a negative effect on casein bioactivities. Although AGEs are the most investigated products from the Maillard reaction, their relationship with intestinal inflammation is still poorly investigated and remains unclear at present [22]. It was thus addressed here that special attention should be focused on the products generated from the early stage of the Maillard reaction (including the glycated proteins), considering their unrevealed bioactivities on the body.

It was confirmed that casein glycation of the TGase-type endowed casein hydrolysates with a greater ability to combat against LPS-induced IEC-6 cell inflammation [60] and enhance and protect the barrier integrity of IEC-6 cells [61,62], while that of the Maillard-type caused casein hydrolysates with an impaired capacity to alleviate LPS-induced IEC-6 cell inflammation (as the present study revealed) and enhance and protect the barrier function of IEC-6 cells [27,34]. Considering that the two types of protein glycation led to different changes in the resultant hydrolysates, it is thus necessary to clarify whether other bioactivities of the hydrolysates are affected by the conducted glycation modification.

## 5. Conclusions

The results of this study highlight that the investigated Maillard-type casein lactose-glycation adversely affect the anti-inflammatory activity of casein hydrolysate in the intestine, considering its capacity to combat against LPS-induced IEC inflammation. Although the glycated casein hydrolysate was capable of alleviating LPS-induced IEC-6 cell inflammation, it perhaps had an ability to cause cellular inflammation and was always less efficient than casein hydrolysates to promote the secretion of anti-inflammatory mediators or to reduce the secretion of pro-inflammatory mediators. It is thus regarded that the assessed casein glycation of the Maillard-type had a negative effect on the anti-inflammatory activity of casein in the intestine, while the lactose-glycation of casein was classified as detrimental to casein biofunction. Whether the protein glycation of this Maillard-type has other positive or negative impacts on protein bioactivities in the body thus needs more investigation in future.

## Figures and Tables

**Figure 1 nutrients-14-05067-f001:**
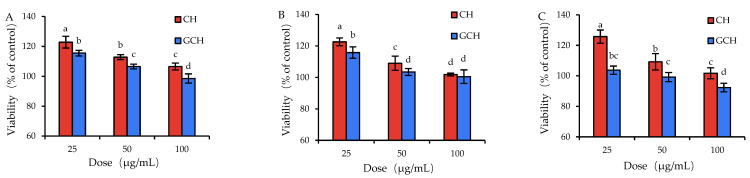
Viability values (%) of the IEC-6 cells treated with casein hydrolysate (CH) and glycated casein hydrolysate (GCH) for 12 h (**A**), 24 h (**B**), and 48 h (**C**), respectively. The one-way ANOVA of the mean values with different lowercase letters above the columns indicates a significant difference (*p* < 0.05).

**Figure 2 nutrients-14-05067-f002:**
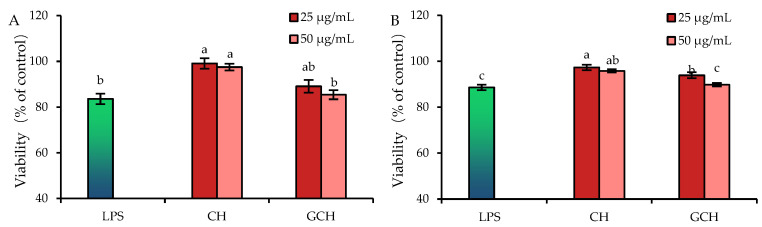
Viability values (%) of the IEC-6 cells treated with casein hydrolysate (CH) and glycated casein hydrolysate (GCH) for 12 h (**A**) and 24 h (**B**), followed by 10 μg/mL LPS treatment for 24 h. The one-way ANOVA of the mean values with different lowercase letters above the columns indicates a significant difference (*p* < 0.05).

**Figure 3 nutrients-14-05067-f003:**
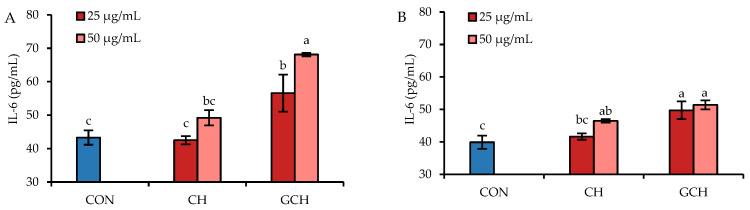
IL-6 secretion in IEC-6 cells treated with casein hydrolysate (CH) and glycated casein hydrolysate (GCH) for 12 h (**A**) and 24 h (**B**). The abbreviation “CON” stands for the control cells without sample treatment, while the one-way ANOVA of the mean values with different lowercase letters above the columns indicates a significant difference (*p* < 0.05).

**Figure 4 nutrients-14-05067-f004:**
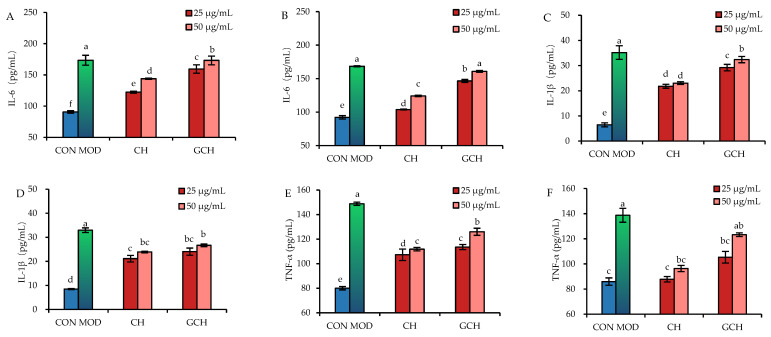
Secretion of IL-6 (**A**,**B**), IL-1β (**C**,**D**), and TNF-α (**E**,**F**) in IEC-6 cells treated with casein hydrolysate (CH) and glycated casein hydrolysate (GCH) for 12 h (**A**,**C**,**E**) and 24 h (**B**,**D**,**F**), respectively, followed by 10 μg/mL LPS treatment of 24 h. The abbreviation “CON” and “MOD” denotes the respective control and model cells. The one-way ANOVA of the mean values with different lowercase letters above the columns indicates a significant difference (*p* < 0.05).

**Figure 5 nutrients-14-05067-f005:**
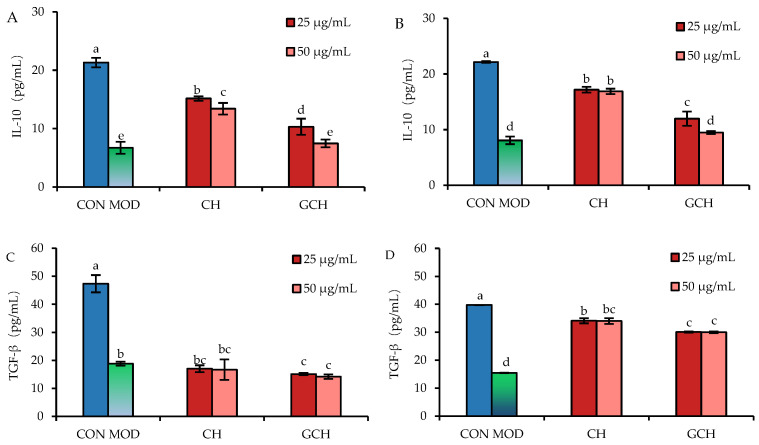
Secretion of IL-10 (**A**,**B**) and TGF-β (**C**,**D**) in IEC-6 cells treated with casein hydrolysate (CH) and glycated casein hydrolysate (GCH) for 12 h (**A**,**C**) and 24 h (**B**,**D**), respectively, followed by 10 μg/mL LPS treatment of 24 h. The abbreviation “CON” and “MOD” denotes the control and model cells, respectively. The one-way ANOVA of the mean values with different lowercase letters above the columns indicates a significant difference (*p* < 0.05).

**Figure 6 nutrients-14-05067-f006:**
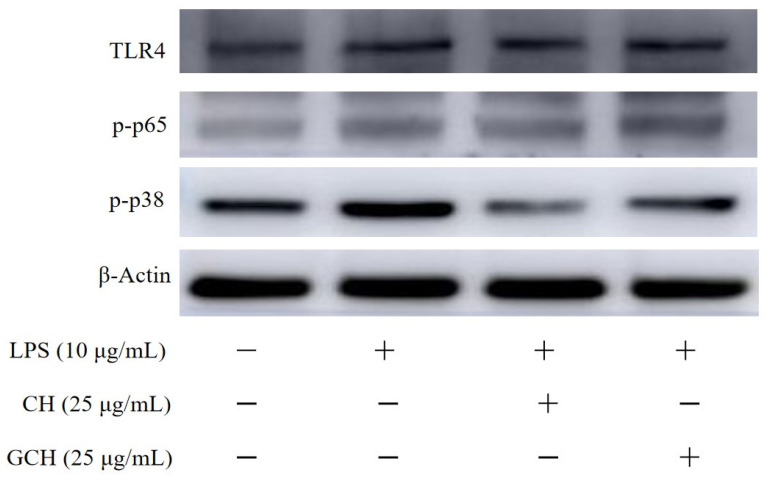
Western blotting assay results of the three proteins TLR4, p-p65, p-p38 in the IEC-6 cells treated with or without casein hydrolysate (CH) and glycated casein hydrolysate (GCH) at a dose of 25 μg/mL for 24 h or LPS treatment (10 μg/mL) of 24 h.

**Table 1 nutrients-14-05067-t001:** The calculated ratios of TLR4:β-actin, p-p65:β-actin, and p-p38:β-actin in the IEC-6 cells treated with casein hydrolysate (CH) and glycated casein hydrolysate (GCH) at 25 μg/mL for 24 h, followed by a LPS treatment (10 μg/mL) of 24 h.

	CON	MOD	CH	GCH
TLR4:β-actin	0.58 ± 0.02	0.75 ± 0.03	0.74 ± 0.05	0.88 ± 0.02
p-p65:β-actin	0.53 ± 0.02	0.60 ± 0.02	0.57 ± 0.04	0.73 ± 0.05
p-p38:β-actin	0.73 ± 0.01	1.01 ± 0.05	0.50 ± 0.01	0.74 ± 0.01

The abbreviation “CON” and “MOD” denotes the control and model cells, respectively.

## Data Availability

All data are contained within the article.

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
