# Peer review of "Casein Lactose-Glycation of the Maillard-Type Attenuates the Anti-Inflammatory Potential of Casein Hydrolysate to IEC-6 Cells with Lipopolysaccharide Stimulation"

_nutrients, 2022, doi:10.3390/nu14235067_

Round 1

Reviewer 1 Report

This manuscript deals with the effect of casein lactose-glycation on the anti-inflammatory potential of casein in the intestine. The work is well presented and can attract the attention of the scientific community, the content is very similar to previous work performed by the same authors and the novelty of this work is not clear. Indeed the authors should highlight the novelty of this work with respect to reference no. 29.

This paper can be considered for publication in Nutrients after some revisions:

Material and methods

Section 2.3. to understand the methodology, we have to search on the references indicated and for the lactose content, the reference indicated doesn't describe the methodology and indicates another reference again. The reader needs to consult several papers to understand the methodology employed. I suggest that the methodology shroud be well described in the present manuscript with clarity. 

Results

Section 3.1. these results are already described in previous work (reference 29) since the same concentrations were tested before using the same approach. However, the results are pretty different. The methodology is not reproducible. The authors should explain this.

Figure 1. Which are the differences with the results of fig 2 of reference 29? You change the nomenclature (codes) of the samples CH- CN hydrolysates (which are the differences)

Figure 2 A (the legend is incomplete, include 50ug/mL)

LPS was added after 24h for both? in which figure are represented A and B or just B? please clarify the figure description.

Lines 276-277: these results are significant?

Figures 3 and 4: Attention to the sample codes. CN and GCN are from previous work. Also Fig 6.

Figure 4D: include 50 ug/mL

Section 3.4. MGCH???

Section 3.5: same as described in previous work (reference 29)

4. Discussion

This section can be reduced. The first paragraphs of this section, until line 396, don't discuss the results obtained in this study.

Lines 406-414: these results are not new.

Lines 441-445: the authors should relate the results with those obtained previously and the bibliography.

Lines 471-473: other products... Such as? 

Conclusion

Lines 475-481: based on these results, these conclusions are not completely true because in some cases are not significant. The authors should take some caution with these types of general conclusions.

Author Response

Point-to-point reply to the comments from the Reviewer #1

This manuscript deals with the effect of casein lactose-glycation on the anti-inflammatory potential of casein in the intestine. The work is well presented and can attract the attention of the scientific community, the content is very similar to previous work performed by the same authors and the novelty of this work is not clear. Indeed the authors should highlight the novelty of this work with respect to reference no. 29.

Reply: Thanks for your suggestion.

Yes, the present study bears some conclusion similarity to the previous one. However, the previous study focused on the role of glycated casein hydrolysate in intestinal barrier enhancement, while the present study highlights the role of glycated casein hydrolysate on the induced inflammation in IEC-6 cells.

The authors agree with your suggestion, and address the novelty of the present work as well as the previous one (previous reference no. 29 but present reference no. 27). Please see our revision in the lines 110-114.

Thanks again!

This paper can be considered for publication in Nutrients after some revisions:

Reply: Thanks! We corrected this manuscript, aiming to give the readers more clear description about the significance of this study.

Material and methods

Section 2.3. to understand the methodology, we have to search on the references indicated and for the lactose content, the reference indicated doesn't describe the methodology and indicates another reference again. The reader needs to consult several papers to understand the methodology employed. I suggest that the methodology shroud be well described in the present manuscript with clarity.

Reply: A valuable comment.

We had added the assaying method for lactose content; please see the revisions in the lines 173-178 in the revised manuscript, with a citation of original reference work.

The first author (Na Chen) of this manuscript prepared these samples independently, and used a kit method to detect lactose content. Thus, we should cite the correct reference but not the present one.

Thanks for your valuable suggestion.

Results

Section 3.1. these results are already described in previous work (reference 29) since the same concentrations were tested before using the same approach. However, the results are pretty different. The methodology is not reproducible. The authors should explain this.

Reply: Thanks for this kindly comment.

(1) The previous work used CCK-8 method to detect call viability, and applied four dose levels of 12.5-100 mg/mL. The results suggested that the 25 mg/mL hydrolysate dose caused the highest viability values.

(2) The present work used MTT method to detect call viability, and applied three dose levels of 25-100 mg/mL. The 25 mg/mL hydrolysate dose also was proven to cause the highest viability values.

(3) As you see, the two different methods were used in the two works, while a similar conclusion was obtained. Thus, the conclusion was reproducible.

(4) However, different viability values were observed, which might be due to the method differences. We do not give detailed discussion here.

(5) Moreover, the used samples were prepared by two authors separately, which a slight difference in lactose content was observed. Thus, a viability analysis was necessary for the performed cell experiments, in our personal opinion.

Very thanks!

Figure 1. Which are the differences with the results of fig 2 of reference 29? You change the nomenclature (codes) of the samples CH- CN hydrolysates (which are the differences)

Reply: Thank you for the kindly comments.

(1) Different authors prepared the samples. Thus, it was necessary to detect the impact of the hydrolysate samples on the model cells.

(2) The present and previous results showed consistent conclusion, although the values of cell viability were not the same (due to different methods were used).

(3) Yes, the authors used new sample codes in this manuscript. The authors of this manuscript prefer using the CH and GCH to represent casein hydrolysate and glycated casein hydrolysate, respectively. As you see, the reference work had used the CN digest and GCN digest to represent casein hydrolysate and glycated casein hydrolysate, respectively. The present sample codes were also acceptable to the readers, in our personal opinion.

Thanks again!

Figure 2 A (the legend is incomplete, include 50ug/mL).

Figures 3 and 4: Attention to the sample codes. CN and GCN are from previous work. Also Fig 6.

Figure 4D: include 50 ug/mL.

Section 3.4. MGCH???

Reply: Sorry, we made some writing mistakes here, although the most errors occurred during a conversion of WORD file to PDF file. Our reply to the 4 related issues is given as below.

As you see, the dose level of 50 μg/mL in Figure 2A and Figure 4D was not fully shown in the PDF version. This error occurred during the conversion of WORD version to PDF one. We have revised this error.

Also, the sample code MGCH was wrong and thus has been revised to GCH.

The full-text has been re-checked carefully and these errors were corrected.

LPS was added after 24h for both? in which figure are represented A and B or just B? please clarify the figure description.

Reply: Thanks!

IEC-6 cells were cultured in medium containing CH or GCH for 12-24 h, and then both were exposed to LPS for 24 h. For doubts, we amended the language in the figure notes, aiming to give the readers more clear result description.

Thanks for your valuable advice.

Lines 276-277: these results are significant?

Reply: Thanks for your suggestion.

The sentences described in the previous lines 276-277 were intended to show that the cells treated with GCH had an inflammatory response, because IL-6 secretion was enhanced clearly. Some previous works had indicated that AGEs had an ability to cause cellular inflammation. Based on this fact, it was suggested a future investigation about possible inflammatory activity of GCH might be necessary.

We had given a minor revision for these sentences. Please see the lines 293-295.

Section 3.5: same as described in previous work (reference 29)

Reply: No, it is not true.

The previous work explored the effect of the samples on cell barrier function, while the present one assessed the effect of the samples on cellular inflammation.

The previous work explored the expression of tight proteins in the cells, but not the proteins involved in signaling pathway. As you see, the present one analyzed the proteins involved in cell inflammation.

WB is widely used to detect protein expression level; thus, a similar description was observed in two manuscripts.

To give the readers critical evidence about anti-inflammatory activities of the samples, these description words or sentences are necessary. Moreover, we also made necessary revisions for this section. Please see the lines 362-378.

  1. Discussion

This section can be reduced. The first paragraphs of this section, until line 396, don't discuss the results obtained in this study.

Reply: Thanks for your kindly suggestion.

The first paragraph of the discussion section aimed to show the readers the background information about the protein glycation as its impact on several protein properties. The second paragraph of the discussion section (till the line 396) was used to show the readers some adverse impacts of the Maillard reaction on protein biofunctions.

As you see, the present study aimed to reveal an adverse influence of the early Maillard reaction on anti-inflammatory activity of casein hydrolysate. Thus, we discussed our results in the second paragraph. If we discussed our results in the first paragraph, it might not be accepted to the readers.

In addition, to give the readers an important background of the Maillard reaction might also be suitable in this manuscript.

We have also revised the two paragraphs, and added our in vivo results that showed another adverse impact of the casein glycation on casein bioactivity in the intestine (the cited reference no. 46). Please see the revised words marked in red.

Thanks again!

Lines 406-414: these results are not new.

Reply: Thanks!

The original sentences in the lines 406-411 of this manuscript focused on our results that supported a decreased anti-inflammatory effect of GCH than CH. In our personal opinion, to obtain the mentioned conclusion (i.e. a negative effect of the assessed casein lactose-glycation on casein bioactivity in the intestine), these results are necessary; meanwhile, these results were in consistence with the published results.

Thanks!

Lines 441-445: the authors should relate the results with those obtained previously and the bibliography.

Reply: Thanks for your suggestion.

We have added more discussion description here. Please see the added sentences in the lines 470-478.

Lines 471-473: other products... Such as?

Reply: Sorry, it was an unclear expression.

We had corrected this expression. Please see our revision in the lines 505-507. Thanks!

Conclusion

Lines 475-481: based on these results, these conclusions are not completely true because in some cases are not significant. The authors should take some caution with these types of general conclusions.

Reply: Thanks for your suggestion.

We have rewritten this section in the resubmitted manuscript. Please see the revised conclusion section in the lines 509-520.

Reviewer 2 Report

In the submitted manuscript, the authors addressed the influence of glycation via the Maillard reaction of casein with lactose on the anti-inflammatory activity. They used an LPS-induced intestinal cell in culture as an inflammation model and measured several cytokine levels.

The issue is essential for a broad audience and involves different science fields: biochemistry to food processing and bioactive substances and clinical problems.

To be accepted for publication, minor but essential revision is recommendable:

1- Consider including the common names of the plants Ganoderma atrum, Arctium lappa, etc, and vice-versa, i.e., Latin names of other cited plants and foods.

2- How are the authors sure that the casein was glycated and that the hydrolysis preserved the glycation? That is, was it experimentally confirmed - the glycation

3- The introduction is well written, but the material & methods section needs a careful revision of the English language and sentence coherence.

4- The discussion section:  It is recommendable that the authors emphasize the difference between the data presented in the submitted article with the data in their previously published reports (refs. 32 and 45).

5-Check the insert values in Figure 2.

6- The legend of figure 3 is confusing, and the definition of the abbreviation MGCH is missing.

Author Response

Point-to-point reply to the comments from the Reviewer #2

In the submitted manuscript, the authors addressed the influence of glycation via the Maillard reaction of casein with lactose on the anti-inflammatory activity. They used an LPS-induced intestinal cell in culture as an inflammation model and measured several cytokine levels. The issue is essential for a broad audience and involves different science fields: biochemistry to food processing and bioactive substances and clinical problems. To be accepted for publication, minor but essential revision is recommendable:

  • Consider including the common names of the plants Ganoderma atrum, Arctium lappa, etc, and vice-versa, i.e., Latin names of other cited plants and foods.

Reply: The authors thanks for your suggestion.

We compared the names of plants and foodstuffs appearing in the text with those in the referenced articles, and thus made necessary corrections in the revised manuscript.

  • How are the authors sure that the casein was glycated and that the hydrolysis preserved the glycation? That is, was it experimentally confirmed - the glycation

Reply: A valuable suggestion.

Protein glycation of the Maillard- and TGase-types is main studying topic of our research group. Protein glycation was proven by SDS-PAGE analyses with both protein and saccharide stainings. We also have measured saccharide levels in the glycated protein by both HPLC and chemical methods, and identified several conjugation sites with HPLC/MS/MS analyses. Property changes of these glycated proteins were also evaluated.

Based on these results, we have published more than 20 papers in the international journals. We do not list these papers here.

Thanks!

  • The introduction is well written, but the material & methods section needs a careful revision of the English language and sentence coherence.

Reply: Thanks!

We paid our best effort to correct this manuscript, aiming to ensure its writing quality. Please see these works marked with red.

  • The discussion section: It is recommendable that the authors emphasize the difference between the data presented in the submitted article with the data in their previously published reports (refs. 32 and 45).

Reply: Thanks for your suggestion.

We had discussed the differences between the present results and previously published results. Please our revised sentences in the lines 418-423.

  • Check the insert values in Figure 2.

Reply: Sorry, an error occurred during preparation of PDF file, which resulted in the loss of hydrolysate dose of 50 μg/mL.

We had paid an attention for this error and then corrected it accordingly.

  • The legend of figure 3 is confusing, and the definition of the abbreviation MGCH is missing.

Reply: Sorry, we made some writing mistakes for these legends.

The sample codes have been revised to CH and GCH. Thanks for your kindly suggestion.

Reviewer 3 Report

The ms “Casein lactose-glycation of the Maillard-type attenuates the anti-inflammatory potential of casein hydrolysate to IEC-6 cells with lipopolysaccharide stimulation” presented by Chen et al. evaluates the anti-inflammatory potential of a glycated casein hydrolysate in lipopolysaccharide trated  rat small intestinal epithelial cells IEC-6.

The overall aim of the paper is to test if the presence of Maillard modified proteins in the intestinal tract promote o prevent a cellular response and if this cellular response if present beneficial or not. 

The presented experimental setup is very straightforward, but the obtained results are very simplistic in the sense that thy give a general "statitical" indication of the cellular response upon the exposure of glycolated casein hydrolysate. 

The data set is a good starting-point for a more exhaustive series of experiments that should define clearly define the protein-lactose conjugates and th cellular response and even think about the use of "whole" animal model. 

Authors should check the body text, figures and figure legends spelling. 

The grafical representation cof bar charts should be consistent. For comparison author might present experimental results in tables and show normalized data in the bar charts. That allows authors to use the same scale for Figures 4 and 5. 

Figure 6: Except for p-p65, protin band intensities are saturated and not suitable for cuantification. Little variance observed for p-p65 can not be excluded to be present in actin control. If data presented in table 1 is calculated from immunoblots similar to Figure 6, then variance should be traeted with caution. 

Author Response

Point-to-point reply to the comments from the Reviewer #3

The ms “Casein lactose-glycation of the Maillard-type attenuates the anti-inflammatory potential of casein hydrolysate to IEC-6 cells with lipopolysaccharide stimulation” presented by Chen et al. evaluates the anti-inflammatory potential of a glycated casein hydrolysate in lipopolysaccharide trated rat small intestinal epithelial cells IEC-6. The overall aim of the paper is to test if the presence of Maillard modified proteins in the intestinal tract promote o prevent a cellular response and if this cellular response if present beneficial or not. The presented experimental setup is very straightforward, but the obtained results are very simplistic in the sense that thy give a general "statitical" indication of the cellular response upon the exposure of glycolated casein hydrolysate.

Reply: The authors thank the reviewer for her/his valuable comment to our study.

The data set is a good starting-point for a more exhaustive series of experiments that should define clearly define the protein-lactose conjugates and th cellular response and even think about the use of "whole" animal model.

Reply: Thanks for your valuable suggestion.

Yes, the aim of this study was to identify whether the mentioned glycation caused activity change in IEC-6 cells. Unfortunately, our results showed that this glycation was adverse to casein, as it caused lower activity to combat against the LPS-induced cell inflammation even possessed an ability to induce cell inflammation.

Whether casein obtained a covalent glycation has been assessed in previous studies; please see the cited reference works and our reply to the comments of the Reviewer 2

We also agree that in vivo investigation for the glycated hydrolysate is necessary, if we obtain future funding support. Furthermore, to obtain more evidence, other cell lines could be considered. Due to our previous results focused on the IEC-6 cells, this study thus employed this cell model again!

Thanks for your valuable advice.

Authors should check the body text, figures and figure legends spelling.

Reply: Sorry, we have made some writing mistakes in the original manuscript.

We have carefully re-checked the full-text and made necessary correction on these writing errors. Please see those words marked with red.

The grafical representation cof bar charts should be consistent. For comparison author might present experimental results in tables and show normalized data in the bar charts. That allows authors to use the same scale for Figures 4 and 5.

Reply: Sorry, we had to use the same scale for the same index but different scales for different indices. This might ensure a more clear presentation about the collected data.

We can use the two indices shown in Figure 4 for a comparison. The measured levels of IL-6 were about 100-200 pg/mL, while those of IL-1b were near 5-40 pg/mL. If the same scale was used to generate the IL-6/IL-1b pictures, the practical values of IL-1b were very unclear to the readers.

Thanks for your kindly suggestion.

Figure 6: Except for p-p65, protin band intensities are saturated and not suitable for cuantification. Little variance observed for p-p65 can not be excluded to be present in actin control. If data presented in table 1 is calculated from immunoblots similar to Figure 6, then variance should be traeted with caution.

Reply: Thanks for your comment.

The protein bands shown in Figure 6 display the results of only one WB assay, whereas the data given in Table 1 are the combined results of three WB assays. We have declared in the Section 2.8. Statistical analysis that “all data collected from at least three independent experiments or assays were ….”.

The band of beta-actin showed the highest intensity. The bands of other proteins thus were regarded unsaturated, in our personal opinion.

Moreover, the expression of p-p65 protein was detected to have 0.57-0.73 fold changes in response to the LPS or hydrolysate exposure. In other words, the treated cells had 1.08-1.34 fold up-regulation for p-p65 protein, compared with the control cells.

Round 2

Reviewer 3 Report

After the revision, the writing of the ms significantly improved, but the experimental data set is the same. Therefore the ms is essentially the same. This set-up is necessary to obtain preliminary results for a more directed study that give rise to strong set of results that sustain the conclusions.